# Deficiency in Dipeptidyl Peptidase-4 Promotes Chemoresistance Through the CXCL12/CXCR4/mTOR/TGFβ Signaling Pathway in Breast Cancer Cells

**DOI:** 10.3390/ijms21030805

**Published:** 2020-01-26

**Authors:** Shaolan Li, Yang Fan, Asako Kumagai, Emi Kawakita, Munehiro Kitada, Keizo Kanasaki, Daisuke Koya

**Affiliations:** 1Department of Diabetology & Endocrinology, Kanazawa Medical University, Uchinada, Ishikawa 920-0293, Japan; shaolanl@kanazawa-med.ac.jp (S.L.); yang_fan@swmu.edu.cn (Y.F.); a-kumagai@juntendo.ac.jp (A.K.); kawakita@kanazawa-med.ac.jp (E.K.); kitta@kanazawa-med.ac.jp (M.K.); 2Division of Anticipatory Molecular Food Science and Technology, Medical Research Institute, Kanazawa Medical University, Uchinada, Ishikawa 920-0293, Japan; 3Internal Medicine 1, Shimane University Faculty of Medicine, 89-1 Enya-cho, Izumo, Shimane 693-8501, Japan

**Keywords:** DPP-4, EMT, chemoresistance, ABC transporters, apoptosis, breast cancer

## Abstract

Dipeptidyl peptidase (DPP)-4, a molecular target of DPP-4 inhibitors, which are type 2 diabetes drugs, is expressed in a variety of cell types, tissues and organs. DPP-4 has been shown to be involved in cancer biology, and we have recently shown that a DPP-4 inhibitor promoted the epithelial mesenchymal transition (EMT) in breast cancer cells. The EMT is known to associate with chemotherapy resistance via the induction of ATP-binding cassette (ABC) transporters in cancer cells. Here, we demonstrated that deficiency in DPP-4 promoted chemotherapy resistance via the CXCL12/CXCR4/mTOR axis, activating the TGFβ signaling pathway via the expression of ABC transporters. DPP-4 inhibition enhanced ABC transporters in vivo and in vitro. Doxorubicin (DOX) further induced ABC transporters in DPP-4-deficient 4T1 cells, and the induction of ABC transporters was suppressed by either the CXCR4 inhibitor AMD3100, the mTOR inhibitor rapamycin or a neutralizing TGFβ (1, 2 and 3) antibody(N-TGFβ). Knockdown of snail, an EMT-inducible transcription factor, suppressed ABC transporter levels in DOX-treated DPP-4-deficient 4T1 cells. In an allograft mouse model, however, the effects of DOX in either primary tumor or metastasis were not statistically different between control and DPP-4-kd 4T1. Taken together, our findings suggest that DPP-4 inhibitors potentiate chemotherapy resistance via the induction of ABC transporters by the CXCL12/CXCR4/mTOR/TGFβ signaling pathway in breast cancer cells.

## 1. Introduction

Diabetes is associated with an increased risk of diverse cancers, and diabetic medicine may influence cancer biology [1,2,3,4]. In particular, diabetic patients need to take diabetic medicine for a long period of time and establishing a safety profile of diabetic drugs is essential for diabetic research. The dipeptidyl peptidase (DPP)-4 inhibitor increases glucagon-like peptide (GIP-1) and promotes insulin secretion in type 2 diabetic patients [5]. However, DPP-4 cleaves many growth factors, chemokines and neuropeptides that may promote human malignancies, and DPP-4 inhibitors potentially increase these substances [6,7,8,9].

There are controversial discussions regarding the biology of DPP-4 in malignancies such as human glioma [7], prostate cancer [10], melanomas [11] and nonsmall cell lung cancer (NSCLC) [12]. C-X-C motif chemokine 12 (CXCL12), the substrate of DPP-4, and its receptor CXCR4 have been associated with the biology of cancer, such as tumor proliferation, survival, invasion and angiogenesis [13]. In this context, DPP-4 could cleave/inactivate CXCL12 and downregulate the CXCL12/CXCR4 axis, subsequently inhibiting cancer cell growth, invasiveness and metastasis [14,15]. Furthermore, a study suggested that DPP-4 might be involved in increased sensitivity of epithelial ovarian carcinoma cells to chemotherapy [16]. We have recently shown that a DPP-4 inhibitor accelerated the epithelial mesenchymal transition (EMT) and lung metastasis via the CXCL12/CXCR4/mTOR axis in breast cancer cells [17]. Signaling through mTOR is vital for proliferation and survival in cancers and plays a central role in adaptive resistance to inhibitors of oncogenic signaling pathways [18,19]. Transforming growth factor-β (TGFβ) signaling, as a tumor suppressor, inhibits cell proliferation but blocks chemotherapy sensitivity. Recent studies have presented evidence that TGFβ signaling acts as a novel antidrug therapeutic target in breast cancer [20] and colorectal cancer [21].

Worldwide, breast cancer is the most common cancer and the leading cause of cancer death in women [22]. Chemotherapy is widely used to treat breast cancer patients [23], and most cancer-related deaths are linked to resistance to chemotherapy. Multidrug resistance (MDR) has been considered as a major determinant in cancer chemotherapy [24,25,26]. ATP-binding cassette (ABC) transporters are the largest family of transmembrane proteins and play a vital role in the development of multidrug resistance and chemoresistance [27] by removing a wide array of commonly employed chemotherapeutic drugs from cancer cells [28]. The overexpression of ABC transporter genes can induce chemoresistance in several cancers [29,30,31]. Recently, researchers focused on the subset of ABC transporters such as P-glycoprotein (also known as ABCB1 or P-gp), MDR-associated protein 1 (MRP1; also known as ABCC1) and ABCG2 (also known as breast cancer resistance protein, BCRP) [32]. The EMT is related to cancer drug resistance and contributes to metastasis after chemotherapy treatment [33]. Regarding breast cancer, cells undergoing the EMT overexpress ABC transporters [34] and are associated with cellular resistance to drug-induced apoptosis [35]. Here, we hypothesize that DPP-4 inhibition increases ABC transporters in the presence of chemotherapy to promote drug resistance in breast cancer.

## 2. Results

### 2.1. Chemotherapy in DPP-4-Deficient Breast Cancer Cells Facilitated the Expression of ABC Transporters

To explore the effects of DPP-4 suppression on chemoresistance in breast cancer cells, we utilized the DPP-4 inhibitor KR62436 or DPP-4 knockdown by specific shRNA (DPP-4-kd) in 4T1 cells. Western blot analysis revealed that the expression of P-gp and ABCG2 had no obvious change in KR-treated 4T1 cells compared with that of the control group. The levels of P-gp and ABCG2 were significantly increased in 4T1 cells treated with doxorubicin (DOX) and were further increased in cells treated with KR and DOX in combination (Figure 1A). DPP-4-kd 4T1 cells exhibited similar DOX-induced P-gp and ABCG2 protein expression (Figure 1B). In MCF-7 cells, KR promoted the expression of ABCG2 and MRP1 either with or without DOX (Figure 1C). A similar trend was observed in the highly metastatic human breast cancer cell line MDA-MB-231; the combination of KR with DOX induced the highest expression levels of ABC transporters (Figure 1D).

### 2.2. DPP-4 Deficiency Induced ABC Transporters via CXCL12/CXCR4/mTOR and the TGF-β Signaling Axis in 4T1 Cells

In our previous study, we have confirmed that DPP-4 inhibition induced the EMT through the CXCL12/CXCR4/mTOR axis in breast cancer [17]. Here, we demonstrated that KR+DOX-induced overexpression of ABC transporters were significantly suppressed by the CXCR4 inhibitor AMD3100 in 4T1 cells (Figure 2A) and were associated with the suppression of CXCL12 and CXCR4 (Appendix A). KR+DOX stimulated the expression of TGFβR1, TGFβR2, smad3 and mTOR phosphorylation in 4T1 cells; AMD3100 suppressed these KR+DOX-induced effects (Appendix A). Similarly, the mTOR inhibitor rapamycin suppressed KR+DOX-induced ABC transporters (Figure 2B) and TGFβR1, TGFβR2 and smad3 phosphorylation (Appendix A). To investigate the role of elevated TGF-β signaling in DPP-4 deficiency-induced CXCL12/CXCR4/mTOR-dependent ABC transporters, we employed neutralizing TGFβ (1, 2 and 3) antibodies (N-TGFβ). Western blot analysis revealed that N-TGFβ blocked the phosphorylation of smad3 and suppressed snail, P-gp and ABCG2 without suppressing mTOR phosphorylation in KR+DOX-treated 4T1 cells (Figure 2C) and MDA-MB-231 cells (Figure 2D).

### 2.3. The Role of EMT in DPP-4 Deficiency-Induced ABC Transporters 

EMT-related transcription factors (TFs) have been shown to play vital roles in regulating ABC transporter expression [36]. As shown previously [17], DPP-4 deficiency induced the EMT, which was characterized by the suppression of E-cadherin and the induction of α-SMA in 4T1 cells, and the trends were much more prominent in KR+DOX treated cells (Figure 3A) and DOX-treated DPP-4-kd 4T1 cells (Figure 3B). Immunofluorescence analysis revealed that primary tumors of DPP-4-kd 4T1 cells exhibited EMT markers (Figure 3C). Snail is a key transcription factor that induces the EMT. Snail knockdown by specific siRNA (siRNA snail) diminished snail levels, resulted in the suppression of the EMT (restoration of E-cadherin and suppression of α-SMA) and was associated with the suppression of ABC transporters P-gp and ABCG2 in 4T1 cells (Figure 3D). However, E-cadherin knockdown-induced EMT did not reverse AMD-3100 suppression of ABC transporters and smad3 phosphorylation in KR+DOX-treated cells (Figure 3E), suggesting that the EMT-inducible transcription factor snail was required for ABC transporter expression induced by DPP-4 deficiency and DOX-treatment; the EMT, as indicated by the suppression of the E-cadherin and the induction of α-SMA, was not sufficient to induce ABC transporters under the same conditions.

### 2.4. The Effects of DPP-4 Deficiency on Chemotherapy-Induced Apoptosis in Breast Cancer Cells

To validate that DPP-4 deficiency-induced ABC transporters were relevant to chemotherapy resistance, we performed an apoptotic assay. DOX and docetaxel (DOC) induced early apoptosis in 4T1 cells, as revealed by an annexin V assay; the proportion of early apoptotic cells was significantly reduced in cells treated with KR combined with either DOX or DOC (Figure 4A,B). As expected, N-TGFβ diminished the KR-induced chemoresistance, suggesting that N-TGFβ sensitized the cells to chemotherapy (Figure 4C), as described previously [20]. KR significantly diminished DOX-induced cleavage of caspase-3 (Figure 4D). Such suppressive effects of KR on the induction of caspase-3 cleavage in DOX-treated cells were diminished by N-TGFβ, AMD3100 and rapamycin (Figure 4E–G).

### 2.5. DPP-4 Deficiency Induced the Expression of ABC Transporters and Was Associated With Chemoresistance in the Allograft Breast Cancer Model

Finally, we tested whether DPP-4 deficiency in tumors was associated with chemoresistance in vivo. DPP-4-kd 4T1 cells displayed accelerated tumor growth when compared to that of shRNA-control 4T1 (control) tumors. DOX significantly suppressed tumor growth in both control and DPP-4-kd 4T1 tumors, but DOX-mediated suppression was less trend in DPP-4-kd 4T1 tumors (Figure 5A; weight suppression rate (%) by DOX: control 42.8% vs. DPP-4-kd 29.7%). DPP-4-kd 4T1 tumors exhibited increased expression of P-gp, ABCG2 and MRP1 in primary tumors compared with that of control tumor-bearing mice, and this trend was enhanced in the presence of DOX (Figure 5B and Appendix A).

Bouin staining of the lung revealed that DPP-4-kd 4T1 tumor-bearing mice exhibited more lung metastasis when compared to control mice with or without DOX; DOX treatment in both control and DPP-4-kd 4T1 tumor-bearing mice displayed some trend of induction in lung metastasis due to an extremely higher incidence of lung nodules in some mice (Figure 5C). However, the difference was not reached significance and was not obvious as well.

## 3. Discussion

Patients with diabetes and cancer have a poor prognosis after treatment with chemotherapy or surgery with a high mortality compared with those without diabetes [2,3]. Chemotherapy is the primary treatment choice for metastatic patients, and chemoresistance is associated with metastasis in breast cancer [37]. Chemoresistance is associated with the EMT. Based on our previous report showing the potential of EMT and metastasis-promoting effects of DPP-4 inhibitors, which are widely prescribed drugs to treat diabetes, we tested whether DPP-4 inhibitors could induce chemoresistance in breast cancer cells. 

Cancer cells undergo EMT, resulting in an enhanced capacity for invasion, metastasis and chemotherapy resistance [38] that is associated with the induction of ATP-binding cassette (ABC) transporters, which are responsible for diverse drug resistances [39]. Ricardo et al. demonstrated that in embryonic development phase, ABC transporters were required for managing the export of a germ cell attractant during directional cell migration in *Drosophila* [40]. Chemoresistance in cancer cells was associated with the EMT and upregulation of ABC transporters could be an evolutionarily conserved system. The overexpression of MDR1 could be conducive to both initiation and acceleration of the chemotherapy resistance in breast cancer cells [41]. Saxena M. et al suggested that chemotherapeutic drug-induced EMT increased the expression of ABC transporters and induced both the drug resistance and the invasion in breast cancer cells [34]. In recent studies, researchers have revealed that overexpression of EMT-related transcription factors (TFs), such as snail, enhanced the chemoresistance through the induction of P-gp and ABCG2 in breast cancer cells [42,43]. In our previous study, we demonstrated that deficiency in DPP-4 induced EMT in 4T1 cells, MCF-7 cells and MDA-MB-231 cells and accelerated lung metastasis of breast cancer in vivo via the CXCL12/CXCR4/mTOR axis [17]. This signaling pathway could be also relevant in the chemoresistance associated with ABC transporters in DPP-4-deficient cells, and EMT-related TFs played a major role in regulating ABC transporters expression, as described in a recent study. Snail has shown to be a major contributor of the maintenance of malignancy potentials and facilitates cancer metastasis and increases chemoresistance [44]. DPP-4 deficiency breast cancer cells treated with chemotherapy promoted the expression of snail. Our in vitro mouse and human cell lines analysis further supports the rationale of DPP-4 deficiency-associated chemoresistance and certain suggestions for clinical relevance.

When compared to the clearer in vitro data, in vivo data is somewhat puzzling. In our studies, DOX suppressed tumor growth in both control and DPP-4-kd 4T1 tumors, but DOX-mediated suppression was less trend in DPP-4-kd 4T1 tumors (tumor weight (g) suppression rate (%): control 42.8% vs. DPP-4-kd 29.7%), but not yet significant. DOX treatment in both control and DPP-4-kd 4T1 tumor-bearing mice displayed an insignificant increase in lung metastasis due to a variation in each sample by an extremely higher incidence of lung nodules in some mice by DOX treatment (either control or DPP-kd). The reason why such difference in each mouse observed was not identified yet. It is reasonable that there is the threshold for ABC transporter to gain chemoresistance; ABC transporter levels in DPP-4-kd tumor in some mice could not be sufficient enough to reach the threshold to induce chemoresistance. Other chemoresistance mechanisms also would be required in this set of experiment in vivo. DPP-4 displays diverse enzymatic and nonenzymatic action. Therefore, DPP-4-kd absolutely induced ABC transporters; parallelly also DPP-4-kd could be associated with the induction of unknown anti-chemoresistance molecular mechanisms in vivo. To explain the discrepancy between in vitro and in vivo data required further investigations.

## 4. Materials and Methods

### 4.1. Reagents and Antibodies

AMD3100 (A5602), KR62436 hydrate (KR, K4264), rapamycin (R8781), doxorubicin (PHR1789) and docetaxel (01885) were obtained from Sigma. The following antibodies were purchased from Abcam: monoclonal rabbit anti-P-glycoprotein (1:1,000, ab170904), monoclonal rabbit anti-ABCG2 (1:1000, ab207732), monoclonal mouse anti-MRP1 (1:1000, ab32574), rabbit polyclonal anti-αSMA (1:1000, ab5694), rabbit polyclonal anti-TGFbR2 (1:1000, ab61213), monoclonal rabbit anti-CD26, (1:1000, ab28340), polyclonal goat anti-CXCR4 (1:1000, ab1670), polyclonal rabbit anti-CXCL12 (1:1000, ab18919) and monoclonal rabbit anti-phospho-Smad3 (s423 and s425; 1:1000, ab84177). The following antibodies were purchased from Cell Signaling Technology: monoclonal rabbit anti-snail (1:1000, 3879S), monoclonal rabbit anti-cleaved-caspase3 (1:1000, CSTD175), anti-caspase3 (1:1000, CST9662S), rabbit polyclonal anti-phospho-mTOR (1:1000, 2971S), rabbit monoclonal anti-mTOR (1:1000, CST2983), rabbit polyclonal anti-Smad3 (1:1000, CST9513), anti-mouse-IgG HRP-linked antibody(1:2000, 7076S), anti-rabbit-IgG HRP-linked antibody(1:2000,7074S) and anti-rat-IgG HRP-linked antibody(1:2000, 7077S). The rat monoclonal anti–E-cadherin antibody was purchased from GeneTex (1:2000, GTX11512). The mouse monoclonal anti–β-actin (1:10,000, A2228) and the rabbit polyclonal anti-TGFBR1 (1:500, SAB1300113) were purchased from Sigma. The neutralizing TGFβ (1, 2 and 3) antibody (MAB1835) was purchased from R&D Systems. Fluorescein (FITC)-conjugated and Cy-conjugated anti-rabbit IgG (1:200, 111-166-047), anti-rat IgG (1:200,112-095-003) and anti-mouse IgM (1:200, A21426) secondary antibodies were obtained from Jackson Immuno Research. Mounting medium containing DAPI (H-1200) was obtained from Vector Laboratories. The annexin V-FITC apoptosis staining/detection kit (ab14085) was purchased from Abcam.

### 4.2. Cell Culture and Treatment

The experimental cell lines, mouse 4T1 breast cancer cells (CRL-2539; GFP-expressing cells), human MCF7 breast cancer cells (HTB-22) and human MDA-MB-231 breast cancer cells (HTB-26), were purchased from ATCC. All cell lines were incubated at 37 °C in a 5% CO_2_ atmosphere, used for 10 passages after reviving from the frozen vials and were regularly stained with DAPI (Vector Labs) to test for *Mycoplasma* contamination every 3 months. 4T1 cells were cultured in RPMI 1640 medium (ATCC) with 10% (final concentration) FBS. MDA-MB-231 cells were cultured in RPMI 1640 medium (ATCC) with 20% FBS. MCF7 cells (HTB-22) were cultured in ATCC-formulated Eagle minimum essential medium with 0.01 mg/mL human recombinant insulin and 10% FBS. The cells were preincubated with KR (50 μmol/L). When the cells reached 70–80% confluence, doxorubicin (DOX) (0.425 μmol/L), docetaxel (DOC) (0.9 μmol/L), AMD3100 (30 μmol/L), the neutralizing TGFβ (1, 2, 3) antibody (N-TGFβ) (1.0 μg/mL) and rapamycin (1 μmol/L) were used to treat the breast cancer cells.

### 4.3. Transfection Experiments

The mouse 4T1 cell DPP-4 shRNA vectors were constructed in the pSIH vector (System Biosciences) using synthetic oligonucleotides against mouse DPP-4 antisense sequences (shDPP-4-1: 50-TAGAAGGAGTATTCAATGAGC-30, shDPP-4-2: 50-AATAGTCAGCTA-GTGAATACG-30 or shDPP-4-3: 50-ATAGTAGAGGATATTTCTTGG-30) with a 50-CTCGAG-30 loop (detailed in a previous work) [17]. 4T1 cells were transfected with siRNA (100 nmol/L) targeting mouse snail (5′-UUGGGAAGUUGGCCCAAAGCCAGGGA-3′; 375453F10, Invitrogen, Tokyo, Japan) and mouse E-cadherin (5′-CGCCACAGAUGAUGGUUCACCCAUU-3′; 379261A08, Invitrogen, Tokyo, country). Based on the manufacturer’s instructions, the cells were incubated using Lipofectamine 2000 (Invitrogen) and siRNA in reduced serum medium for 6 h. Then, the medium was exchanged with experimental medium. The 4T1 cells were treated with or without DOX (0.425 μmol/L) for 48 h after snail siRNA transfection.

### 4.4. Allograft Breast Cancer Mouse Model

Eight-week-old female BALB/c mice were obtained from Inc. Japan (CLEA Japan). The DPP-4 knockdown by specific shRNA (DPP-4-kd) or shRNA-control (control) 4T1 cells (5 × 10^5^ cells in 20 μL of PBS) were orthotopically implanted into the mammary fat pads of each mouse using a Hamilton syringe fitted with a 25G needle. Concomitantly, the mice were randomly allocated to one of the following four groups: (1) control; (2) DPP-4-kd-4T1; (3) control +DOX and (4) DPP-4-kd-4T1+DOX. When the tumor volumes reached 80–100 mm^3^, the mice were intraperitoneally injected with DOX (5 mg/kg, once a week). Tumor sizes were measured on every alternate day with a digital caliper, and volumes were calculated using the following formula: tumor volume (mm^3^) = (width^2^) × (length/2). To quantify the tumor growth rate, we determined the relative tumor volume compared to the average volume of the starting tumor (0 day) for each group. Twenty-one days after treatment with DOX, the mice were sacrificed, and the primary tumors and lungs were removed and analyzed. The experiments described herein were executed in line with the animal protocols of Kanazawa Medical University (lentiviral shRNA in vivo experiment protocol number 2018-17).

### 4.5. Ethics Statement

The Animal Use Committee of Kanazawa University approved all animal study protocols (2019-15; 2019-06-17), and all experiments were conducted in accordance with the guidelines for the care and use of laboratory animals.

### 4.6. Western Blot Analysis

Western blot analysis was performed according to standard protocols. Total proteins were extracted by RIPA lysis buffer with PMSF, sodium orthovanadate and a protease inhibitor cocktail (Santa Cruz Biotechnology). Then, the protein lysates were boiled in SDS sample buffer at 94 °C for 5 min, separated on SDS-polyacrylamide gels, and transferred to PVDF membranes (Pall Corporation, Pensacola, FL, USA) using the semidry method. After being blocked with TBS-T (Tris-buffered saline containing 0.05% TWEEN 20) containing 5% non-fat dry milk, the membranes were incubated with each primary antibody at 4 °C overnight, followed by incubation with the appropriate secondary antibody for 1 h at room temperature. The signal was developed with an enhanced chemiluminescent substrate detection solution, and the membranes were imaged using an ImageQuant LAS 4000 mini (GE Healthcare Life Sciences, Uppsala, Sweden).

### 4.7. Flow Cytometry

The cells were seeded in a 6-well plate (1 × 10^6^ cells/well) and preincubated with KR (K4264) (50 μmol/L). When the cells reached 70–80% confluence, they were treated with DOX (0.425 μmol/L) or DOC (0.9 μmol/L). The incubator was maintained at 5.0% CO_2_ and 37 °C. After 24 h of treatment, the cells were washed with cold PBS three times, trypsinized using trypsin-EDTA (Invitrogen) and centrifuged at 2000 rpm for 5 min. The cells were resuspended and washed 2 times with cold PBS, centrifuged at 2000 rpm for 5 min and resuspended in banding buffer, and then annexin V-FITC was added. Subsequently, the cells were incubated at room temperature for 5 min in the dark. Cell fluorescence was measured by flow cytometry (Gallios, Beckman Coulter, Tokyo, Japan).

### 4.8. Immunofluorescence in Mouse Tissue

The frozen tumor sections were fixed with 4% paraformaldehyde phosphate buffer solution for 30 min at 4 °C. Then, the sections were washed twice with PBS and blocked with 2% BSA/PBS for 30 min at room temperature. Then, the tissues were incubated with primary antibody overnight at 4 °C, washed with PBS and incubated with the corresponding secondary antibody for 30 min. The tissues were then gently washed and shaken three times with PBS and mounted with mounting medium containing DAPI. The images were analyzed by fluorescence microscopy (BZ-X710 Viewer, KEYENCE, Osaka, Japan).

### 4.9. Bouin Buffer Staining

Bouin buffer solution was prepared as follows: 10% formaldehyde: 0.9% picric acid:5% acetic acid at ratios of 15:5:1. First, the lung specimens were perfused with 10% formaldehyde and then fixed in Bouin solution for at least 24 h after dissection. Subsequently, surface lung metastasis was quantitated by counting the number of metastatic nodules according to the Bouin staining.

### 4.10. Statistical Analysis

The data were analyzed utilizing one-way analysis of variance, followed by Tukey’s multiple comparison test to confirm statistical significance, which was considered *p* < 0.05, unless otherwise noted. GraphPad Prism software (ver. 7.0f; La Jolla, CA) was utilized for the statistical analysis. The figures display the means and standard deviations (mean ± s.e.m).

## 5. Conclusions

In conclusion, we described that (1) DPP-4 deficiency induced ABC transporters in mouse and human breast cancer cell lines; (2) the transcription factor snail was required for DPP-4-deficiency-induced ABC transporters and E-cadherin suppression with α-SMA induction was not essential; (3) the inhibition of the CXCR4/mTOR axis or TGF-β signaling sensitized 4T1 cells with DPP-4 deficiency to chemotherapy-induced apoptosis and (4) in an allograft model, the effects of DOX in either primary tumor or metastasis were not statistically different between control or DPP-4-kd 4T1 even though some mice displayed extremely high prevalence of metastasis in DOX-treated DPP-4-kd tumor bearing mice. These data further demonstrated the significance of DPP-4 inhibition in the biology of cancer. Diabetes is known to have a higher incidence of cancer, and clinicians are required to treat diabetes during chemotherapy. Clinical evidence related to this topic requires further evaluation; one may be aware of choosing the proper diabetic medicine for cancer-bearing patients, especially patients with chemoresistance and CXCR4-positive cancers.

## Figures and Tables

**Figure 1 ijms-21-00805-f001:**
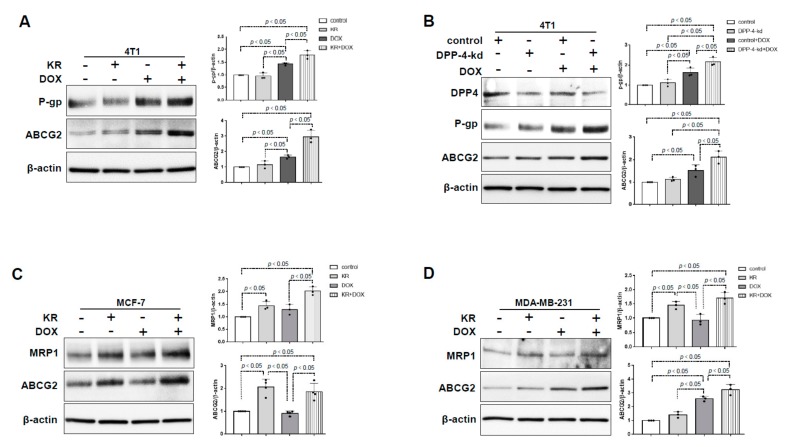
DPP-4 suppression enhanced the expression of ATP-binding cassette (ABC) transporters in breast cancer cells treated with chemotherapy drugs. (**A**) Western blot analysis of P-gp and ABCG2 in 4T1 cells pretreated with KR62436 (50 μmol/L) for 48 h and then treated with or without doxorubicin (DOX;0.425 μmol/L) for another 48 h. (**B**) Western blot analysis of P-gp and ABCG2 in DPP-4 shRNA knockdown (DPP-4-kd) and shRNA control (control) 4T1 cells treated with or without DOX (0.425 μmol/L) for 48 h. (**C**,**D**) Western blot analysis of MRP1 and ABCG2 in MCF-7 cells (**C**) and MDA-MB-231 cells (**D**) pretreated with KR62436 (50 μmol/L) for 48 h and subsequently treated with or without DOX (0.425 μmol/L) for another 48 h. All representative blots from three independent experiments were shown, and densitometric analysis of protein expression relative to β-actin levels was performed by using ImageJ.

**Figure 2 ijms-21-00805-f002:**
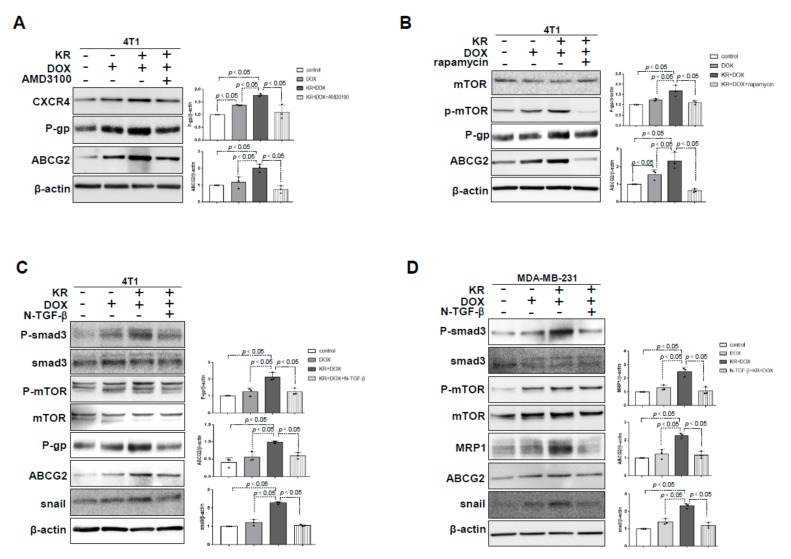
DPP-4 inhibition increased ABC transporters in combination with chemotherapeutic insult in a CXCL12/CXCR4/mTOR/TGF-β-dependent manner in breast cancer cells. (**A**,**B**) Western blot analysis of P-gp and ABCG2 in 4T1 cells pretreated with KR62436 (50 μmol/L) for 48 h and then treated with DOX (0.425 μmol/L) in the presence or absence of AMD3100 (30 μmol/L) (**A**) and rapamycin (1 μmol/L; **B**) for another 48 h. (**C**,**D**) Western blot analysis of mTOR phosphorylation (P-mTOR), P-gp, ABCG2 and snail in 4T1 cells (**C**) and MDA- MB-231 cells (**D**) pretreated with KR62436 (50 μmol/L) for 48 h and then treated with DOX (0.425 μmol/L) in the presence or absence of a neutralizing TGFβ (1, 2 and 3) antibody (N-TGFβ, 1.0 μg/mL) for another 48 h. All representative blots from three independent experiments were shown, and densitometric analysis of protein expression relative to β-actin levels was performed using ImageJ.

**Figure 3 ijms-21-00805-f003:**
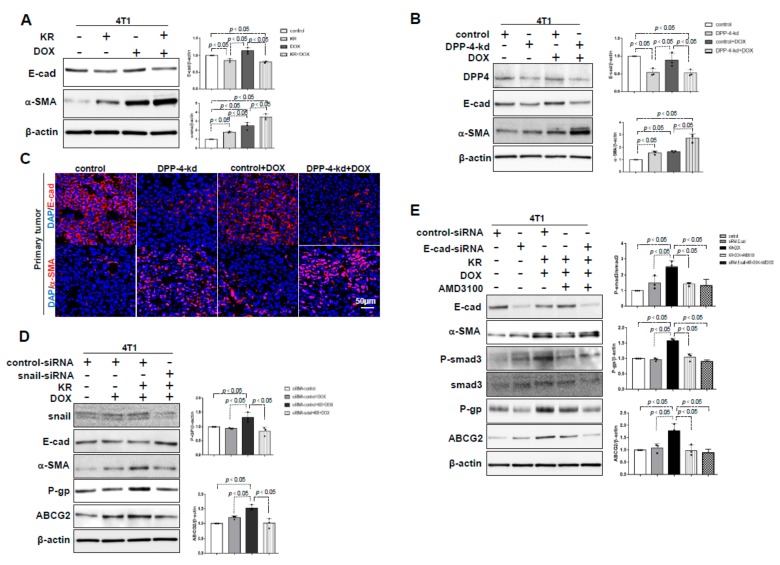
DPP-4 inhibition-induced epithelial mesenchymal transition (EMT) impacted ABC transporters during chemotherapy treatment of breast cancer. (**A**) Western blot analysis of E-cadherin and α-SMA in 4T1 cells pretreated with KR62436 (50 μmol/L) for 48 h and then treated with or without DOX (0.425 μmol/L) for another 48 h. (**B**) Western blot analysis of E-cadherin and α-SMA in control and DPP-4-kd 4T1 cells treated with or without DOX (0.425 μmol/L) for 48 h. (**C**) Immunofluorescence analysis of E-cadherin and α-SMA expression in DPP-4-kd and control primary tumors treated with or without DOX. For each group, representative images of six different fields of view at 200× magnification were evaluated. The scale bar indicates 50 μm in each panel. (**D**) Western blot analysis of P-gp and ABCG2 in snail siRNA knockdown (snail siRNA) and control siRNA (100 nmol/L) 4T1 cells pretreated with KR62436 (50 μmol/L) for 24 h and then treated with or without DOX (0.425 μmol/L) for another 48 h. (**E**) Western blot analysis of phosphorylated smad3 (P-smad3), P-gp and ABCG2 in E-cadherin siRNA knockdown (E-cadherin siRNA) and control siRNA (100 nmol/L) 4T1 cells pretreated with KR62436 (50 μmol/L) for 24 h and then treated with or without DOX (0.425 μmol/L) for another 48 h. All densitometric quantification relative to β-actin levels and P-smad3 protein expression relative to smad3 levels (*n* = 3 per group) were performed by using ImageJ.

**Figure 4 ijms-21-00805-f004:**
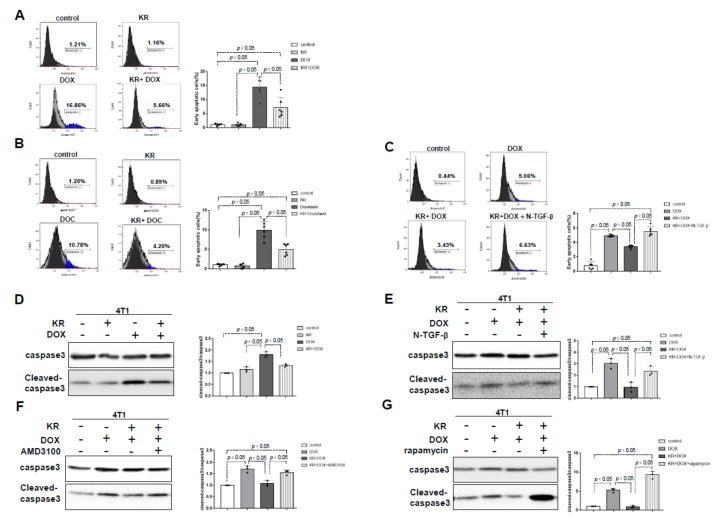
DPP-4 inhibition protects breast cancer cells from apoptosis. (**A**–**C**) Detection of early apoptosis utilizing flow cytometry (annexin V-FITC apoptosis staining) in 4T1 cells pretreated with KR62436 (50 μmol/L) for 48 h and then treated with or without doxorubicin (0.425 μmol/L; **A**) or docetaxel (DOC; 0.9 μmol/L; **B**) for another 24 h in the presence or absence of the neutralizing TGF-β (1, 2 and 3) antibody (N-TGFβ, 1.0 μg/mL; **C**) for another 24 h. Densitometric analysis of early apoptotic cells (%) in each group (*n* = 6 per group). (**D**) Western blot analysis of cleaved caspase-3 in 4T1 cells pretreated with KR62436 (50 μmol/L) for 48 h and then treated with or without DOX (0.425 μmol/L) for another 48 h. (**E**–**G**) Western blot analysis of 4T1 cells pretreated with KR62436 (50 μmol/L) for 48 h and subsequently treated with or without DOX (0.425 μmol/L) in the presence or absence of the neutralizing TGFβ (1, 2 and 3) antibody (N-TGFβ, 1.0 μg/mL; **E**), AMD3100 (30 μmol/L; **F**), or rapamycin (1 μmol/L; **G**) for another 48 h. All densitometric analyses of protein expression relative to the caspase3 levels (*n* = 3 per group) were performed by using ImageJ.

**Figure 5 ijms-21-00805-f005:**
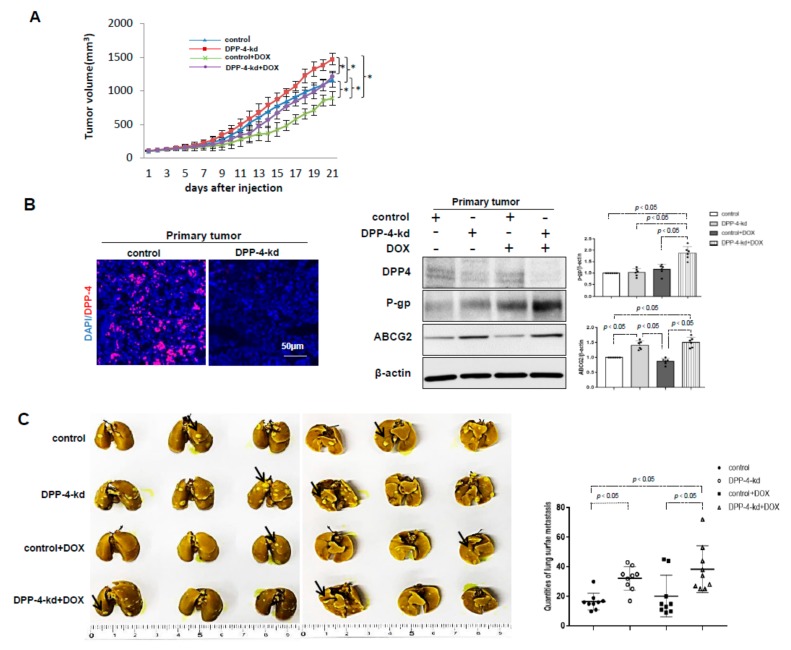
The influence of DPP-4 knockdown on promoting primary tumor growth, metastasis and chemoresistance in vivo. Eight-week-old female BALB/c mice were orthotopically implanted with DPP-4 shRNA knockdown (DPP-4-kd) and shRNA-control (control) 4T1 cells into mammary fat pads of each mouse. Concomitantly, the mice were randomly allocated to one of the following four groups: (1) control; (2) DPP-4-kd; (3) control + DOX and (4) DPP-4-kd+DOX groups. When the tumor volumes reached 80–100 mm^3^, mice were intraperitoneally injected with DOX (5 mg/kg, once a week). Twenty-one days after treatment, the mice were sacrificed, and the primary tumors and lungs were analyzed. (**A**) The tumor volume in each group was measured ever day during treatment (* *p* < 0.05). (**B**) Immunofluorescence analysis of DPP-4 expression in control and DPP-4-kd primary tumors. Western blot analysis of P-gp and ABCG2 expressions in the primary tumor tissue. Densitometric analysis of protein expression relative to β-actin levels (*n* = 6 per group) was performed by using ImageJ. (**C**) The lung surface metastases (left panel) were imaged, and the quantification of lung metastases (right panel) was performed by Bouin staining. Black arrows indicated lung surface metastases. The data in the graphs was shown as mean ± SEM; *n* = 9.

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
