# Peer review of "Deficiency in Dipeptidyl Peptidase-4 Promotes Chemoresistance Through the CXCL12/CXCR4/mTOR/TGFβ Signaling Pathway in Breast Cancer Cells"

_ijms, 2020, doi:10.3390/ijms21030805_

Round 1

Reviewer 1 Report

The paper ”Deficiency in dipeptidyl peptidase-4 promotes chemoresistence...........” by Li et al. is a very interesting approach regarding the chemoresistance of cancer cells in diabetic patients.  

Unfortunately, the authors worked with animals, but no approval of the ethics committee was declared.

The Discussion section must be improved. The authors have to discuss extensively about EMT in breast cancer cells and its correlation with ABC transporters. Also they have to discuss about the lung metastasis. Also the information presented in lines 230-239 is very suitable for conclusions and not at the begining of discussion section.

Author Response

The paper “Deficiency in dipeptidyl peptidase-4 promotes chemoresistence...........” by Li et al. is a very interesting approach regarding the chemoresistance of cancer cells in diabetic patients.  

----- We thank reviewer’s positive evaluation and constructive comments on our manuscript.

Point 1:Unfortunately, the authors worked with animals, but no approval of the ethics committee was declared.

Response 1:Thank you for picking up this. We forgot to state such important information in original submission. In the revised manuscript, we stated that “the Animal Use Committee of Kanazawa University approved all animal study protocols (2019-15), and all experiments were conducted in accordance with the guidelines for the care and use of laboratory animals”.

Point 2:The authors have to discuss extensively about EMT in breast cancer cells and its correlation with ABC transporters. Also they have to discuss about the lung metastasis.

Response 2:We are thankful to reviewer’s superb insight on our paper. In revised manuscript, we added more discussion including the discrepancy between in vivo and in vitro data about chemoresistance as well.

Point 3:The Discussion section must be improved. Also the information presented in lines 230-239 is very suitable for conclusions and not at the beginning of discussion section.

Response 3:We thank reviewer for constructive comments. We modified the discussion section.

Reviewer 2 Report

This study “Deficiency in dipeptidyl peptidase-4 promotes chemoresistance through the CXCL12/CXCR4/mTOR/TGFβ signaling pathway in breast cancer “ by Shaolan Li et al addresses the role of DDP4 in the pathogenesis of breast cancer. This is an interesting topic.

The manuscript is easy to read, and the in vitro experiments are well conducted. By contrast, the in vivo experiments are not convincing with histological panels of bad quality.

In addition, resistance to anticancer drugs, mainly doxorubicin here, is not clearly supported by the authors’ data.

Major concerns:

Figure 2A: DDP4 inhibition in 4T1 cell lines does not alter sensitivity to doxorubicin. It seems more that DDP4 inhibition increases tumor growth after been grafted into syngenic mice. It is associated with an increased expression of efflux pumps, which does not necessary mean chemoresistance. And chemoresistance to which drug? Doxorubicin? That is not the case here. As an oncologist with expertise in pathology, I do not understand the panel D. What do the authors want to show with HE pictures? Immunostainings are unreadable, and pictures are of very bad quality. They are pixelated. I have the same comment for Figure 3 that does not help to bring any conclusion. Figure 3: The “Bouin coloration” to identify and count lung metastases is a debatable approach. Again, the authors are misinterpreting their results: DDP4 inhibition (DDP4-kd) is not only associated with an increase in 4T1 cell proliferation but also an increase in metastatic potential. And it does not seem that doxorubicin treatment increases the numbers of lung metastases in mice grafted with 4T1-DDP4-kd cells. This result would have been surprising and of interest.

Figure 2B is useless and redundant with Fig 2A.

Author Response

This study “Deficiency in dipeptidyl peptidase-4 promotes chemoresistance through the CXCL12/CXCR4/mTOR/TGFβ signaling pathway in breast cancer “ by Shaolan Li et al addresses the role of DDP4 in the pathogenesis of breast cancer. This is an interesting topic.The manuscript is easy to read, and the in vitro experiments are well conducted. By contrast, the in vivo experiments are not convincing with histological panels of bad quality.In addition, resistance to anticancer drugs, mainly doxorubicin here, is not clearly supported by the authors’ data.

-----We thank reviewer’s positive evaluation and constructive comments on our manuscript.

Point 1:Figure 2A: DDP4 inhibition in 4T1 cell lines does not alter sensitivity to doxorubicin. It seems more that DDP4 inhibition increases tumor growth after been grafted into syngenic mice. It is associated with an increased expression of efflux pumps, which does not necessary mean chemoresistance. And chemoresistance to which drug? Doxorubicin? That is not the case here.

Response 1:This is superb point of reviewer. An active efflux mechanism has been believed as the main reason for multidrug resistance in tumor cells treated with chemotherapy. ABC transporters are the largest family of transmembrane proteins which bind ATP and use the energy to drive the drugs across all cell membranes. Therefore, ABC transporters are responsible for diverse drug resistance. In our studies, DOX significantly upregulated ABC transporters (P-gp, ABCG2 and MRP1) levels in vivo and in vitro. Also DOX suppressed tumor growth in both control and DPP-4-kd 4T1 tumors, but DOX-mediated suppression was decreased trend in DPP-4-kd 4T1 tumors (Tumor weight (g) suppression rate (%): control 42.8% vs. DPP-4-kd 29.7%), but not yet significant. We modified description in this data and added the discussion about the discrepancy between in vitro and in vivo data.

Point 2:As an oncologist with expertise in pathology, I do not understand the panel D. What do the authors want to show with HE pictures? Immunostainings are unreadable, and pictures are of very bad quality. They are pixelated.

Response 2:We are thanks for review’s superb insight on our paper. We need to apologize for the confusion. We also asked pathologist in our hospital to evaluate this and she advised us that primary tumor in all groups are similarly too much aggressive phenotype and cannot say which one the most. Therefore, in our revised paper, we removed these HE staining panels. Also for the IHC analysis for ABC transporters, we already showed the western blot analysis data, therefore we also removed them. Now we updated the data in Figure 5.

Point 3:I have the same comment for Figure 3 that does not help to bring any conclusion. Figure 3: The “Bouin coloration” to identify and count lung metastases is a debatable approach. Again, the authors are misinterpreting their results: DDP4 inhibition (DDP4-kd) is not only associated with an increase in 4T1 cell proliferation but also an increase in metastatic potential. And it does not seem that doxorubicin treatment increases the numbers of lung metastases in mice grafted with 4T1-DDP4-kd cells. This result would have been surprising and of interest.

Response 3:We agree with the reviewer’s comment. When compare to the clearer in vitro data, in vivo data is somewhat puzzling. DOX treatment in both control and DPP-4-kd 4T1 tumor-bearing mice displayed an insignificant increase in lung metastasis due to a variation in each sample by an extremely higher incidence of lung nodules in some mice by DOX treatment (either control or DPP-4-kd 4T1). The reason why such difference in each mouse occurs is not identified yet. Maybe there is the threshold for ABC transporter to gain chemoresistance and ABC transporter levels in DPP-4-kd 4T1 tumor in some mice was not sufficient enough to reach the threshold to induce chemoresistance. Other chemoresistance mechanisms also could be required in this set of experiment in vivo. DPP-4 displays diverse enzymatic and nonenzymatic action, therefore alternatively, DPP-4-kd absolutely induced ABC transporters, but also could be associated with the induction of unknown anti-chemoresistance molecular mechanisms in parallel. To explain the discrepancy between in vitro and in vivo data required further investigations.

We moved in vivo data from Figure 2 to Figure 5, since as reviewer suggested that in vitro data was much strait-forward to suggest the chemoresistance by DPP-4 deficiency.

Point 4:Figure 2B is useless and redundant with Fig 2A.

Response 4: We removed the panels.

Round 2

Reviewer 1 Report

In this form, the paper of Shaolan et al. can be accepted for publication in IJMS.

Reviewer 2 Report

The authors have correctly answered to my criticisms, and the manuscript is significantly improved.